# Mapping Quantitative Trait Loci in *Arabidopsis* MAGIC Lines Uncovers Hormone-Responsive Genes Controlling Adventitious Root Development

**DOI:** 10.3390/plants14111574

**Published:** 2025-05-22

**Authors:** Brenda Anabel López-Ruiz, Joshua Banta, Perla Salazar-Hernández, Daniela Espinoza-Gutiérrez, Andrea Alfaro-Mendoza, Ulises Rosas

**Affiliations:** 1Jardín Botánico, Instituto de Biología, Universidad Nacional Autónoma de México, México City 04510, Mexico; brenda.lopez@st.ib.unam.mx (B.A.L.-R.); salazar.p.karina@gmail.com (P.S.-H.); daniespinozagut@gmail.com (D.E.-G.); andrea.alfaro.m28@gmail.com (A.A.-M.); 2Department of Biology and Center for Environment, Biodiversity, and Conservation, University of Texas at Tyler, 3900 University Blvd., Tyler, TX 75799, USA; jbanta@uttyler.edu

**Keywords:** adventitious roots, *Arabidopsis thaliana*, MAGIC lines, natural variation, QTL mapping, root architecture

## Abstract

The Multi-Parent Advanced Generation Inter-Cross (MAGIC) population is a powerful tool for dissecting the genetic architecture controlling natural variation in complex traits. In this work, the natural variation available in *Arabidopsis thaliana* MAGIC lines was evaluated by mapping quantitative trait loci (QTLs) for primary root length (PRL), lateral root number (LRN), lateral root length (LRL), adventitious root number (ARN), and adventitious root length (ARL). We analyzed the differences in the root structure of 139 MAGIC lines by measuring PRL, LRN, LRL, ARN, and ARL. Through QTL mapping, we identified new potential genes that may be responsible for these traits. Furthermore, we detected single-nucleotide polymorphisms (SNPs) in the coding regions of candidate genes in the founder accessions and the recombinant inbred lines (RILs). We identified a significant region on chromosome 1 associated with AR formation. This region encompasses 316 genes, many of which are involved in auxin and gibberellin signaling and homeostasis. We discovered SNPs in the coding regions of these candidate genes in the founder accessions that may contribute to natural variation in AR characteristics. Additionally, we found a novel gene that encodes a protein from the hydroxyproline-rich glycoprotein family, which exhibits differential SNPs in accessions with contrasting AR formation. This study provides genetic insights into the natural variation in AR numbers using MAGIC lines linked to hormone-related genes.

## 1. Introduction

The root system anchors the plant and absorbs water and nutrients; studying root architecture and growth is essential in the field of plant organ development [1]. *Arabidopsis thaliana* (hereafter *Arabidopsis*) has a simple root system, which makes it helpful in studying root architecture and identifying new genes that control these characteristics. Moreover, it is relatively easy to visualize the root system of *Arabidopsis* on vertical agar plates *in vitro* [2].

The *Arabidopsis* root system consists of three main types of roots: the primary root (PR), the lateral roots (LRs), and the adventitious roots (ARs) [3,4]. The PR originates from the embryonic meristem and is the first organ to develop during germination [5]. Its principal function is to anchor the plant into the soil by growing downwards [3]. The LRs develop from founder cells in the pericycle adjacent to the protoxylem poles of the vascular cylinder, and they explore the soil in response to environmental clues [6]. The ARs, on the other hand, develop from non-root organs such as hypocotyl, stems or leaves. They are observed when *Arabidopsis* is grown vertically on a synthetic medium or in response to wounding or environmental signals [4,7]. Several genes have been identified to play a vital role in developing PR, LR, and AR [8,9,10,11]. These genes include transcription factors such as *WUSCHEL-RELATED HOMEOBOX* (*WOX*), *PLETHORA* (*PLT*), *AUXIN RESPONSE FACTORS* (*ARFs*), *SHORT ROOT* (*SHR*), and *SCARECROW* (*SCR*), as well as cell cycle genes like *CYCLIN-DEPENDENT KINASES* and *CYCLINS*. Additionally, plant hormones are crucial regulators of cell division, elongation, and differentiation [8,9,10].

In *Arabidopsis*, the Multi-Parent Advanced Generation Inter-Cross (MAGIC) population is a large set of recombinant inbred lines (RILs) that are used to map a vast amount of phenotypic diversity, combining the benefits of association mapping and linkage analysis; these lines offer great power for detecting quantitative trait loci (QTL) [12]. The MAGIC lines were developed by crossing 19 natural accessions of *A. thaliana* over multiple generations [13]. This breeding strategy not only introduced additional recombination events but also increased the number of segregating alleles within the lines. Consequently, QTLs can be pinpointed to much smaller genomic regions than usual, often less than 1 Mb in size [13]. Moreover, they contain 3.3 million segregating SNPs, which is 68% of SNPs with a frequency above 0.05 identified by the 1001 genome project, thus capturing a substantial portion of the standard molecular variation present in the species [13]. MAGIC populations increase genetic recombination and genetic variation and reduce the limitations of biparental populations for QTL mapping (lack of mapping precision and low genetic diversity) [14]. Potentially, each MAGIC line can inherit alleles from all the progenitors, and the MAGIC chromosomes are random mosaics of the parental haplotypes [14]. The MAGIC population is well-suited for breeding improvement, and analysis of the relationship between genotypes and phenotypes allows the identification of QTLs, which can then be confirmed using functional genomics [15,16].

The *Arabidopsis* MAGIC population has been used to map several traits involved in germination and bolting time [17] and to examine the natural genetic basis of variation in seed size and number [18]. However, the genetic basis of root phenotypic variation in MAGIC lines has not yet been addressed. This study aimed to evaluate the variation in root architecture in 139 recombinant inbred lines. Additionally, QTL mapping was performed to detect new candidate genes contributing to these traits.

## 2. Results

### 2.1. Root Variation in Arabidopsis MAGIC Lines

We analyzed five root characteristics in 139 *Arabidopsis* recombinant inbred lines derived by intermating 19 natural accessions [17]. These 139 MAGIC lines are the populations that we managed to propagate, which directly come from the original work published by Kover et al. (2009) [17]. We aimed to preserve the original populations as intact as possible, ensuring they remain suitable for QTL analysis. We recorded the length of the primary root (PRL), the number of lateral roots (LRN), the length of lateral roots (LRL), the number of adventitious roots (ARN), and the length of adventitious roots (ARL) for 18 days after sowing (DAS) by marking their positions in the Petri dish. The variability in root architecture of each one of the individual values of the 139 MAGIC lines can be observed in Figure 1A–E.

Appendix A presents the mean and standard deviation (SD) of the 139 MAGIC lines; the range of PR varies from 0.22 cm ± 0.28 to 3.38 cm ± 2.9 cm. The LR number goes from 0 to 12.46 ± 15.10, and concerning ARN, some MAGIC lines lack ARs, and the highest value is 3.5 ± 3.53. Regarding ARL, the minimum mean length is 0.019 ± 0.03 cm, whereas the maximum is 8.01 ± 11.2 cm. This considerable SD has been observed previously in other studies that have used the *Arabidopsis* recombinant inbred lines [19].

We conducted multiple correlations to study the association between the five traits using the mean of the individual’s data of each of the 139 MAGIC Lines (Figure 2). The results show the highest positive correlation (0.882) between the PR length and the LR number. We also found strong associations between the number of LRs and the length of ARs and LRs (0.781 and 0.747, respectively). However, the correlation between the number of ARs and the length of LRs was the weakest (0.457) among all (Figure 2). This indicates that some traits are highly related, possibly to physical constraints, such as the length of the PR and the number of LRs.

### 2.2. QTLs Accounted for Adventitious Root Number and Candidate Genes

We conducted QTL mapping through association analysis of the five root traits studied: PRL, LRN, LRL, ARN, and ARL. However, we only obtained reliable results for ARN data. We set the LOD score threshold to >11 to determine significance and identified a significant peak on chromosome 1 in the trait for adventitious root number (ARN) (Figure 3). The physical intervals of the QTL covered 316 genes, most of which were protein-coding genes (Appendix A).

To evaluate whether these 316 genes exhibit polymorphisms in the MAGIC and parental lines, we compared our dataset with that of Kover et al. (2009) [17], which includes a list of 1536 SNPs that cover the entire genome of the founder and MAGIC lines. Our focus was specifically on chromosome one, which contains only 275 polymorphisms (Appendix A). To identify the genes associated with these SNPs, we consulted the TAIR database. We detected that only three SNPs, MN1_18632912, MASC00545, and MN1_19400778, mapped to genes also found within the QTL interval. These genes include AT1G50300, AT1G51140, and AT1G52120, which encode for a *TBP-ASSOCIATED FACTOR 15*, a *BASIC HELIX-LOOP-HELIX-TYPE TRANSCRIPTION FACTOR* involved in photoperiodic flowering, and a *MANNOSE-BINDING LECTIN SUPERFAMILY PROTEIN*. None of these genes has been functionally validated for their role in adventitious root development. We created a categorical heatmap for these three polymorphisms to visualize the SNPs between the founders and the MAGIC lines. This allows us to observe the genetic diversity among the 139 lines used in this study and the 19 parental lines (Figure 4).

Due to the limited number of SNPs available for the MAGIC lines, we conducted a literature review and transcriptome analysis to explore the functions of all 316 genes. While searching through GeoDatasets, we found a transcriptome related to *de novo* adventitious root formation [20]. Of the 316 genes located in the QTL peak, 177 were identified in this transcriptome (Appendix A). Moreover, we identified which genes in our candidate gene list were previously recognized as regulators of AR formation. We found that *TARGET OF RAPAMYCIN* (*TOR*) and *INDOLE-3-ACETIC ACID INDUCIBLE18* (*IAA18*) affect AR initiation [21,22,23]. Furthermore, GA biosynthesis, signaling, and auxin homeostasis are involved in AR formation [24]; therefore, we selected two additional hormone-related genes: *PLETORA* (*PLT2*) and *ARABIDOPSIS THALIANA GIBBERELLIN 2-OXIDASE 7* (*GA2ox7*). *PLT2* has been suggested to participate in AR primordium formation since *plt1 plt2 plt3* triple homozygotes arrest its AR growth and development [25]. In addition, it has been observed that GA hinders the growth of adventitious roots by interrupting auxin transport [26]. While GA2ox7 has not been recognized as a regulator of AR, it could facilitate AR development by reducing GA levels through hydroxylation and deactivation of GA precursors [27]. In addition, it has been reported that specific accessions exhibit differential AR formation when exposed to auxin [28], and these accessions are shared with the *Arabidopsis* founder accession used in this study. Because of this, we selected a gene that encodes for a hydroxyproline-rich glycoprotein family protein that, although it has not yet been characterized, is of interest due to the presence of differential SNPs in these accessions with contrasting AR phenotypes after auxin treatment (medium-high rooting: Bur-0, Edi-0, Ler-0; low rooting: Mt-0, Wil-2, Oy-0, Ct-1) (Appendix A; Table 1).

### 2.3. Allelic Variants in Five Candidate Genes of the Founder Accessions

We searched for allelic variants in the five candidate genes (*TOR*, *GA2ox7*, *PLT2*, *IAA18* and hydroxyproline-rich glycoprotein family protein) of the founder accessions to identify nucleotide changes that may contribute to natural phenotypic variation in ARs observed in the recombinant inbred lines (Appendix A). Our search was limited to the coding region, and we only selected significant changes such as missense variants or in-frame deletions (Table 1). Upon analyzing *TOR*, we found that although there were numerous changes in its 56 exons, they were mainly synonymous variants that did not affect the translated amino acid sequence. The main change was a three-nucleotide in-frame deletion in the Rsch-4 accession localized in the kinase domain (protein position: 2368–2369 amino acid; Figure 5). *PLT2* has two deleterious variants in No-0 and Ct-1 that do not localize within the AP2/ERF domains, whereas Sf-0 and Ct-1 share an in-frame deletion that could shift the gene open reading frame. In addition, *IAA18* has several tolerated missense variants in the AUX/IAA domain and only one deleterious variant in Zu-0 that changes the amino acid from Glycine to Valine in the same domain (Figure 5).

On the other hand, *GA2ox7* has six missense variants, two of which are deleterious in Oy-0 and Hi-0, and whose position is in the Diox-N domain, a highly conserved N-terminal region with 2-oxoglutarate/Fe (II)-dependent dioxygenase activity (Table 1; Figure 5).

The SNP 1:18250388 mapped in the gene that encodes a hydroxyproline-rich glycoprotein family protein; this variation results in a change from Proline to Serine in a coding region that has not been characterized (Figure 5). This SNP is present in accessions like Bur-0, Edi-0, Ler-0, and En-2, which have a high capability of forming AR [28] (Appendix A). However, accessions like Mt-o, Wil-2, An-1, Oy-0, and Ct-1, with low competence of AR formation, do not have this variant (Appendix A. Table 1). In addition, we detected that Can-0, Tsu-0, and Zu-0 accessions contain a harmful genetic variation, whereas Po-0 displays two deleterious SNPs (Table 1).

## 3. Discussion

There has been a longstanding interest in understanding the link between genetic and phenotypic variation in natural populations. This insight is crucial for identifying the genetic basis of adaptation and discovering naturally occurring alleles that influence various traits. The root system is vital to plant growth and productivity. Therefore, it is necessary to comprehend the genetic basis of natural variation in this particular trait [29].

One approach to identify genes responsible for natural variations in complex traits, such as root architecture, is to link genetic and phenotypic differences using recombinant inbred lines. In the case of MAGIC lines, there are several parental accessions and additional recombination generations [17]. Although only 19 accessions were used to establish the MAGIC recombinant inbred lines, they still capture a significant portion of the common genetic variation present in *Arabidopsis* [17]. In this work, we evaluated five root traits: PR length, LR and AR length, and number; we observed a wide range of variations in phenotype among the MAGIC lines for all the measured traits, and a positive correlation between the length of PR and the number of LR was detected. It has previously been reported that there is no correlation between the length of the primary root (PR) and the number of lateral roots in *Arabidopsis* natural accessions [30]. However, we have observed the opposite trend and believe as the lateral roots emerge from the PR, longer PRs may indicate more physical space for lateral root initiation.

QTL mapping revealed a significant region on chromosome 1 associated with ARN. ARs originate post-embryonically from aerial parts such as stems or hypocotyls, and they are induced by many environmental and physiological stresses to expand absorbing areas or enhance resistance to adversity [31,32]. ARs are used in asexual propagation, which might be difficult to promote in many crops. Thus, understanding its molecular mechanisms is essential for such species [33,34]. AR development is a complex process controlled by diverse factors such as phytohormones, particularly auxin. The critical step in AR formation is the development of AR primordia, which begins with the auxin synthesis and accumulation [35]. According to the repositories AraPheno and AraGWAS, no data exist about the genetic associations between distinct loci and AR development. In this work, we detected 316 genes associated with ARN, with some associated with auxin and GA synthesis, signaling, and transport. It has been discovered that genes encoding proteins associated with gibberellin biosynthesis and signaling and auxin homeostasis are involved in forming AR [36].

Auxin and TOR work together to regulate AR formation. TOR is a protein kinase and a master regulator that integrates energy, nutrients, stress, and hormone signaling to promote cell growth and proliferation [37]. When *Arabidopsis* roots are treated with TOR inhibitors, the formation of ARs is significantly slower. However, the overexpression of the auxin receptor TIR1 can partially restore the formation of ARs that were inhibited by TOR inhibitors. This suggests that there may be a connection between the TOR and auxin signaling mechanisms during the AR formation process [21]. In addition, we identified a nucleotide in-frame deletion in the kinase domain of TOR from the parental accession Rsch-4. This deletion might contribute to the natural variation in AR.

Moreover, the gain-of-function mutant *crane-2*, which harbors mutations in domain II of IAA18 that confers resistance to degradation by the proteasome, was also shown to be affected in AR initiation [23]. We observed that IAA18 has several missense variants in the AUX/IAA domain and only one deleterious variant in Zu-0 that leads to an amino acid change. On the other hand, *PLT2* is a crucial gene that plays a significant role in maintaining the identity of the quiescent center (QC), root apical meristem (RAM) maintenance, and activating the gene expression of polar auxin transport, biosynthesis, and response genes [38,39]. It has also been associated with AR initiation [25]. We detected that some parental accessions of MAGIC lines have an in-frame deletion that could shift the gene open reading frame of *PLT2*.

Additionally, for the polymorphism analysis, we selected a gene that encodes for a hydroxyproline-rich glycoprotein family protein that displays differential SNPs in accessions with contrasting AR phenotypes. These accessions come from a study where the variation in the number of ARs formed on seedling hypocotyls in response to auxin was evaluated in 18 ecotypes [28]. Mt-0, Wil-2, and An-1 are accessions with low ARs formed, whereas Bur-0, En-2, Ler-0, and Edi-0 are accessions that display a high number of AR formation after auxin treatment. We noticed that an SNP changes from Proline to Serine in a coding region in these accessions with high AR formation. The cross between Mt-o and Ha-S (accession with high capability of AR formation) shows a low proportion of high rooting segregants in the F2, suggesting a multigene control in AR formation [28].

Regarding GA regulation in AR formation, we noticed that GA2ox7 has six missense variants, two of which are deleterious in Oy-0 and Hi-0, whose position is in the Diox-N domain, a highly conserved N-terminal region with 2-oxoglutarate/Fe (II)-dependent dioxygenase activity. It has been described that loss-of-function mutations of *GA REQUIRING 1* (*GA1*) and *GA5*, which encode for the enzymes ENT-COPALYL DIPHOSPHATE SYNTHETASE 1 and GA 20-OXIDASE, respectively, lead to a significant reduction in the number of ARs in both hypocotyl explants and excised leaves [36]. Likewise, it has been reported that treating WT plants with GA4 (1 μm) significantly inhibited adventitious rooting [26]. Similarly, transgenic plants with higher GA biosynthesis via overexpressing GA20ox1 have significantly fewer ARs in stem cuttings due to the perturbation of polar auxin transport [26].

## 4. Materials and Methods

### 4.1. Growth Conditions and Phenotyping

The MAGIC population used in this study was created with 19 *Arabidopsis* founder accessions (Bur-0, Can-0, Col-0, Ct-1, Edi-0, Hi-0, Kn-0, Ler-0, Mt-0, No-0, Oy-0, Po-0, Rsch-4, Sf-2, Tsu-0, Wil-2, Ws-0, Wu-0, and Zu-0) [17]. A total of 139 lines, each consisting of approximately ten individuals, were used to record the root traits for the study. Seeds were disinfected with a solution of 70% ethanol, 50% bleach, and three rinses of sterilized water. Seeds were sown on square Petri dishes with 50% Murashige and Skoog salts. MS medium includes the following: macronutrients NH_4_NO_3_, KNO_3_, CaCl_2_·2H_2_O, MgSO_4_·7H_2_O, and KH_2_PO_4_, micronutrients H_3_BO_3_, MnSO_4_·H_2_O, ZnSO_4_·7H_2_O, KI, Na_2_MoO_4_·2H_2_O, CuSO_4_·5H_2_O, CoCl_2_·6H_2_O, FeSO_4_·7H_2_O, and Na_2_EDTA, and organic compounds such as myo-inositol, thiamine·HCl, pyridoxine·HCl, and nicotinic acid (MSP09, Caisson, Utah, USA). The medium was supplemented with 0.05% MES and 1% agar (Bacto Agar BD, Difco, Montreal, Canada) at pH 5.6. The plates with the seeds were stratified for three days at 4 °C in darkness and then placed vertically in a growth chamber under 22 °C, long-day 16 h/8 h at 200 µmol/m^2^/s light intensity (Percival Scientific, Iowa, USA). We consider each plant to be an experimental unit, which is nested within a block, being the plate. Preliminary tests indicated no block effect due to the plate where the plant is growing; therefore, the block effect was not considered when estimating parameters. Eighteen days after sowing (das), plates were digitized at 600 dpi using a scanner Epson V600 (Nagano, Japan), and five root traits were measured using the Image J version Fiji software (Version 1.54p, NIH, Maryland, USA) using a ruler as a scale: primary root length, lateral root number, lateral root length, adventitious root number, and adventitious root length.

### 4.2. Data Visualization and Statistical Analysis

For trait visualization, we used box plots to depict the data distribution for each trait. These box plots illustrate the median and the first and third quartiles, with whiskers extending to the minimum and maximum values within 1.5 times the interquartile range. Any data points outside this range are considered outliers and displayed as individual dots above the boxes. Each box plot color corresponds to individual MAGIC lines and was assigned based on their increasing order (left to right) using a manually defined color palette.

Additionally, we performed a correlogram using the ggpairs function from the GGally package, employing the Spearman correlation coefficient to assess the relationships between root traits, since our variables are not normally distributed and include outliers.

### 4.3. QTL Mapping

QTL mapping was performed using the ‘scan1’ function of the R/qtl2 package R, in combination with a custom R data package containing the genotype data in a suitable format for analysis (available at https://github.com/joshbanta/qtl-mapping-magic-lines, modified based on the original script from https://github.com/tavareshugo/atMAGIC, accessed on 1 April 2025). Genome-wide significance was determined empirically for each trait, using 1000 permutations of the data with the ‘scan1perm’ function of the R/qtl2 package, corresponding to a genome-wide false positive rate of 5%. A total of 139 MAGIC lines for five root traits were used. Appendix A shows a step-by-step workflow for QTL mapping and gene overlay that we followed.

### 4.4. Search for Candidate Genes in the QTL Peak in Chromosome 1

The list of candidate genes within the QTL interval was identified using TAIR10, and detailed descriptions of each gene were obtained from ShinyGo 0.82 (https://bioinformatics.sdstate.edu/go/, accessed on 1 December 2023). We assessed whether these genes had been previously identified in transcriptome studies related to adventitious root formation by referencing GeoDatasets (https://www.ncbi.nlm.nih.gov/gds/?term=adventitious+roots+arabidopsis/, accessed on 1 April 2025). The list of genes was then compared with those identified in the QTL interval, and the common genes, along with a scaled matrix of values, were visualized using a heatmap created with the pheatmap R package version 1.0.12.

### 4.5. Polymorphism Patterns

We compared the supplementary dataset from [17], which contains a list of 1536 SNPs covering the entire genome of the founders and MAGIC lines, with the 136 lines that we evaluated. We focused specifically on chromosome one, which includes only 275 polymorphisms. To determine which genes mapped to these SNPs, we searched the TAIR database. We visualized the common genes associated with these polymorphisms and those found within the QTL interval for AR number using a categorical heatmap created with ggplot2.

Sequence data from the 1001 genome project [40] (http://signal.salk.edu/atg1001/3.0/gebrowser.php; accessed on 1 December 2023) were used to analyze single-nucleotide polymorphisms (SNPs) of candidate genes among founder accessions (Bur-0, Can-0, Col-0, Ct-1, Edi-0, Hi-0, Kn-0, Ler-0, Mt-0, No-0, Oy-0, Po-0, Rsch-4, Sf-2, Tsu-0, Wil-2, Ws-0, Wu-0, and Zu-0). To determine the effects of these variations and their exact positions, Variant Effect Predictor (VEP; [41]) was executed on the coding sequence, with default parameters (https://plants.ensembl.org/Arabidopsis_thaliana/Tools/VEP (accessed on 1 December 2023). The prediction of an amino acid substitution effect influences protein function based on sequence homology and the physicochemical similarity between the alternate amino acids. Each amino acid substitution is assigned a score and a qualitative prediction: either ‘tolerated’ or ‘deleterious’. The qualitative prediction is based on this score, where substitutions with a score less than 0.05 are categorized as ‘deleterious,’ while all other substitutions are labeled as ‘tolerated’.

We searched the AraPheno (https://arapheno.1001genomes.org accessed on 1 December 2023) and AraGWAS Catalog (https://aragwas.1001genomes.org accessed on 1 December 2023) data repositories for publicly available phenotypic data and distinct loci related to adventitious root formation. Protein domains were retrieved from InterPro version 105.0, visualized using Unipro UGENE version 52.1, and edited with Adobe Illustrator version 29.5.1.

## 5. Conclusions

In summary, this study evaluated various physical traits related to the root structure of MAGIC lines and identified a significant QTL on chromosome 1 for the number of adventitious roots. This QTL is associated with several genes related to phytohormones, including those involved in auxin and gibberellin signaling and homeostasis, most of which have previously been identified in the transcriptome of novo AR formation. Auxins and gibberellins are relevant hormones that participate in adventitious root formation. Auxins are positive regulators that play a key role in developing adventitious root primordia, whereas gibberellic acid has been described as an inhibitor of adventitious rooting. Additionally, we discovered novel genes linked to polymorphisms in the recombinant inbred lines that have not been previously associated with AR formation. Some founder accessions exhibited missense mutations and in-frame deletions in these genes, which could lead to natural variation in AR development. Further research is needed to fully understand the complex interactions between gibberellin and auxin signaling and their role in the natural variation in AR development.

## Figures and Tables

**Figure 1 plants-14-01574-f001:**
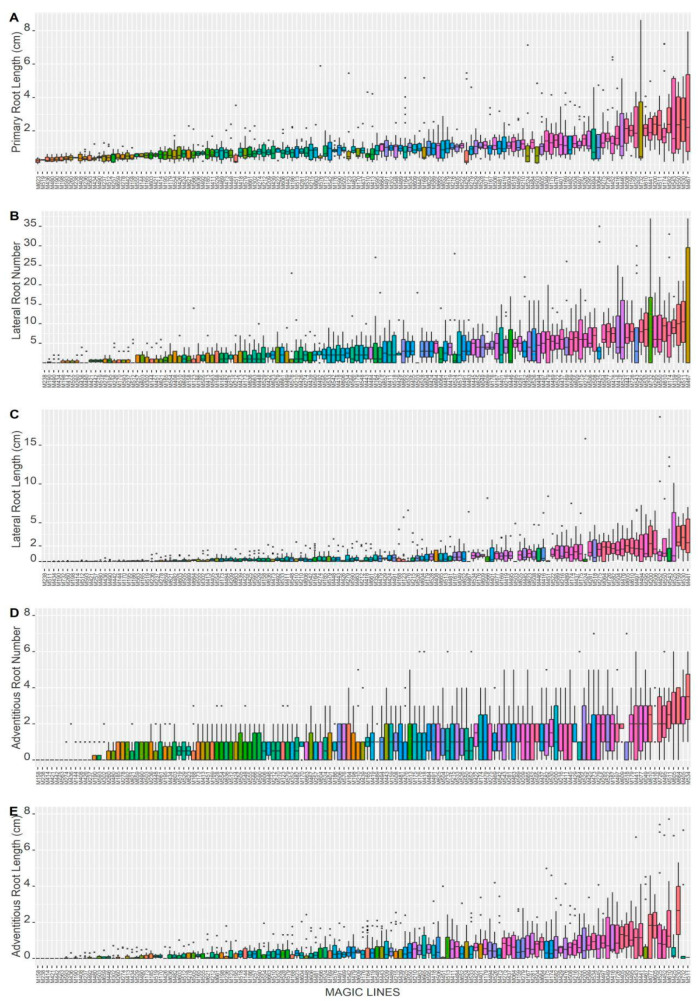
Five root traits were quantified in the *Arabidopsis* MAGIC lines: (**A**) primary root length (PRL), (**B**) lateral root number (LRN), (**C**) lateral root length (LRL), (**D**) adventitious root number (ARN), and (**E**) adventitious root length (ARL). The box plot displays the first and third quartiles, median, and whiskers extending to the maximum or minimum value. The dots above each box plot correspond to outliers, which were identified as data points that fall outside of the range defined by the whiskers. The color associated with each box was obtained using a continuous color palette to differentiate the 139 MAGIC lines.

**Figure 2 plants-14-01574-f002:**
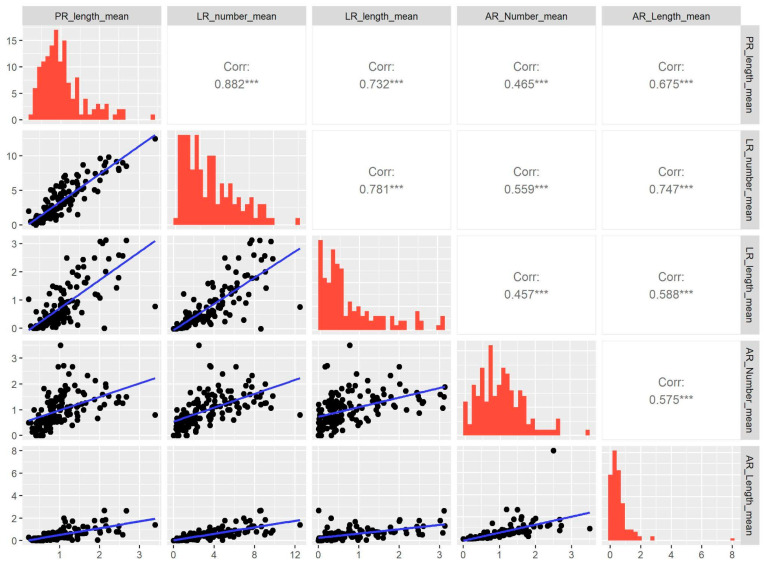
Pair plots indicate the correlation between the five root traits analyzed in this study. The upper panel of the plots exhibits the correlation between the traits, while the lower panel displays the scatter plots of the traits. The histograms are displayed on the diagonal. The significant value is indicated with (***) *p* < 0.001. Primary root (PR), lateral root (LR), and adventitious root (AR). The significance of the correlation was determined using the Spearman correlation coefficient.

**Figure 3 plants-14-01574-f003:**
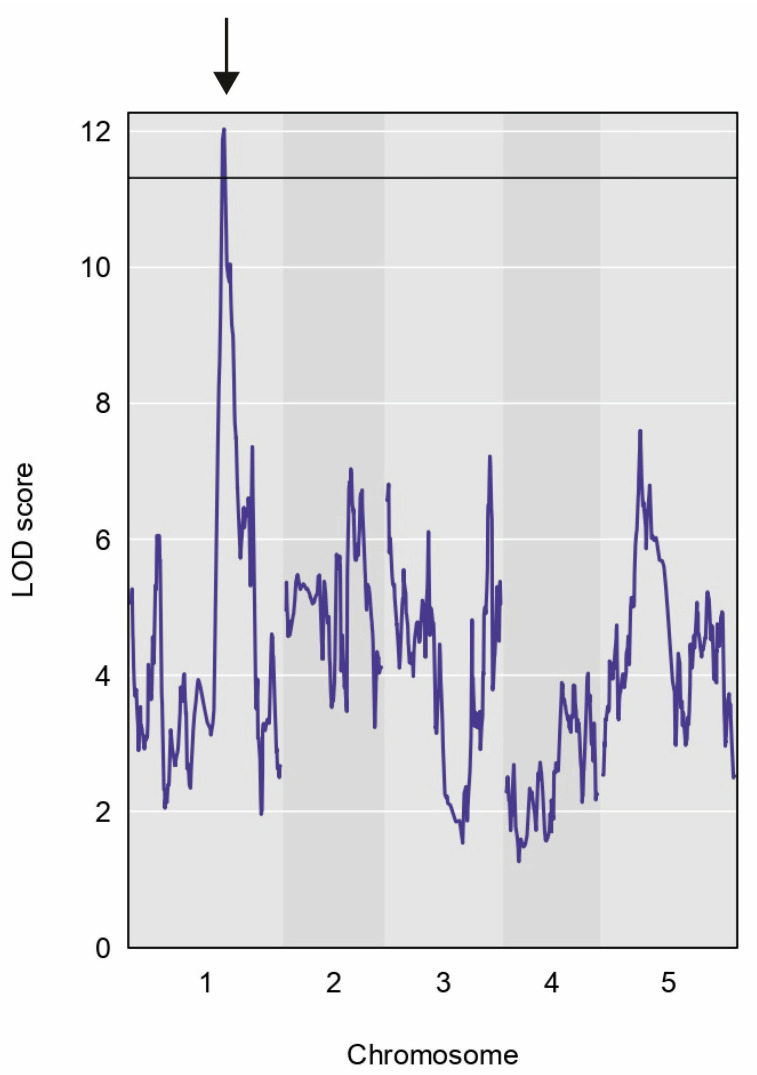
Chromosomal location of significant QTLs for the adventitious root number (ARN); a peak on chromosome 1 (indicated with an arrow) is detected that spans 316 genes.

**Figure 4 plants-14-01574-f004:**
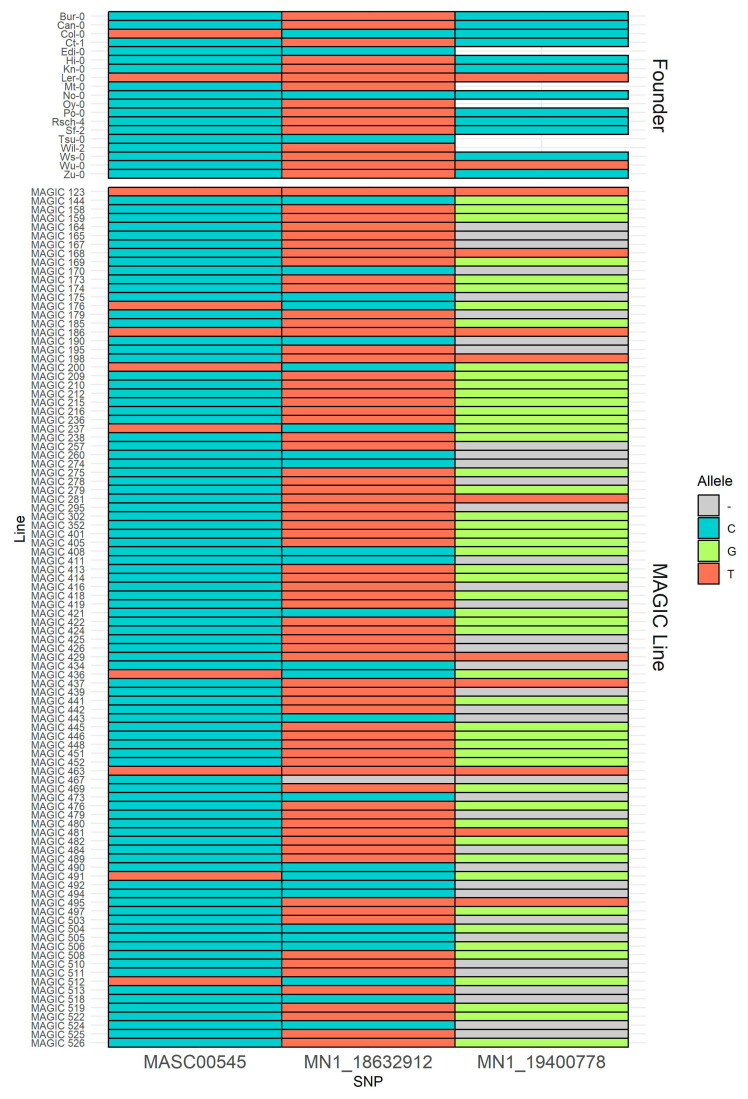
Categorical heatmap of three SNPs: MN1_18632912, MASC00545, and MN1_19400778, which are associated with AT1G50300, AT1G51140, and AT1G52120 within the QTL interval for adventitious root number. This figure illustrates the genetic diversity among the MAGIC lines and parental lines used in this study. A distinct color represents each nucleotide base.

**Figure 5 plants-14-01574-f005:**
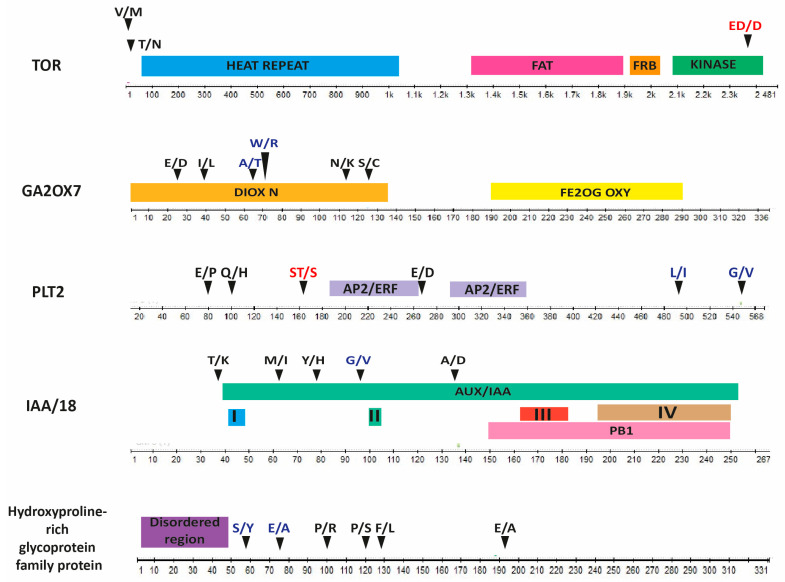
Proteins obtained via QTL mapping and their changes in founder accessions. Each protein domain is represented in different colors. Amino acid changes are displayed with arrowheads. Amino acid deletions are represented in red, deleterious missense variants are in blue, and tolerated changes are in black. HEAT (huntingtin, elongation factor 3 (EF3)), FAT (FRAP, ATM, and TRAP), FRB (FKB12-rapamycin binding), DIOX-N (N-terminal region of proteins with 2-oxoglutarate/Fe (II)-dependent dioxygenase activity), FE2OG_OXY (Fe(^2+^) 2-oxoglutarate dioxygenase domain profile), AP2/ERF (APETALA 2/Ethylene-Responsive Element binding factors), AUX/IAA (AUXIN/INDOLE-3-ACETIC ACID), and PB1 (Phox and Bem1 domain). Also, the four conserved domains of IAA18 are shown. Domain I contains the EAR (ERF-associated amphiphilic repression) motif involved in transcriptional repression. Domain II is essential for interaction with TIR1/AFB proteins in the presence of auxin, leading to ubiquitination and degradation of the protein. Domains III and IV are involved in protein dimerization, allowing the formation of homo- or heterodimers with other Aux/IAA proteins or ARFs.

**Table 1 plants-14-01574-t001:** Candidate genes associated with adventitious root numbers were identified via QTL mapping. The position and impact of the allelic variants of these genes in the coding regions of the *Arabidopsis* founder accessions are indicated. According to the Variant Effect Predictor from Ensembl, each amino acid substitution is assigned a score and a qualitative prediction. A substitution is considered deleterious if its score is less than 0.05, while a high score indicates it is tolerated.

Gene	Founder Accessions	Chr: bp	Alleles	Class	Conseq. Type		AA
At1G50030 TOR	Rsch-4	1:18523147-18523149	TCT/-	Deletion	Inframe deletion	-	ED/D
	Kn-0	1:18539595	C/T	SNP	Missense variant	Tolerated	V/M
	No-0	1:18539612	G/T	SNP	Missense variant	Tolerated	T/N
At1g50960 GA2ox7	Can-0, Hi-0, Kn-0, No-0, Rsch4, Sf-2	1:18889623	G/T	SNP	Missense variant	Tolerated	E/D
	Can-0, Sf-2	1:18889660	A/T	SNP	Missense variant	Tolerated	I/L
	Hi-0	1:18889744	G/A	SNP	Missense variant	Deleterious	A/T
	Oy-0	1:18889756	T/A	SNP	Missense variant	Deleterious	W/R
	Oy-0	1:18889890	T/A	SNP	Missense variant	Tolerated	N/K
	Mt-0	1:18889922	C/G	SNP	Missense variant	Tolerated	S/C
At1G51190 PLT2	Bur-0, Can-0, Ct-1, Edi-0, Hi-0, Kn-0, Mt-0, Oy-0, Po-0, Rsch-4, Sf-2, Tsu-0, Wil-2, Ws-0, Wu-0, and Zu-0	1:18978006	A/C/T	SNP	Missense variant	Tolerated	E/P
	Kn-0	1:18978071	A/T	SNP	Missense variant	Tolerated	Q/H
	Can-0, Sf-0	1:18978257-18978259	CAC/-	Deletion	Inframe deletion	-	ST/S
	Kn-0	1:18979010	G/C	SNP	Missense variant	Tolerated	E/D
	Ct-1	1:18980072	C/T/A	SNP	Missense variant	Deleterious	L/I
	No-0	1:18980238	G/T	SNP	Missense variant	Deleterious	G/V
At1G51950 IAA18	Kn-0	1:19305779	C/A	SNP	Missense variant	Tolerated	T/K
	Mt-0	1:19305855	G/A	SNP	Missense variant	Tolerated	M/I
	Mt-0	1:19305901	T/C	SNP	Missense variant	Tolerated	Y/H
	Zu-0	1:19306234	G/C	SNP	Missense variant	Deleterious	G/V
	Can-0, Hi-0, Kn-0, Ler-0, Mt-0, No-0, Wil-2, Ws-0Zu-0	1:19306355	C/A	SNP	Missense variant	Tolerated	A/D
AT1G49330 Hydroxyproline- rich glycoprotein family protein	Can-0, Po-0, Tsu-0, Zu-0	1:18250209	C/A	SNP	Missense variant	Deleterious	S/Y
	Po-0	1:18250260	A/C	SNP	Missense variant	Deleterious	E/A
	No-0	1:18250338	C/G/T	SNP	Missense variant	Tolerated	P/R
	Bur-0, Ler-0, Edi-0	1:18250388	C/T	SNP	Missense variant	Tolerated	P/S
	No-0	1:18250417	C/A	SNP	Missense variant	Tolerated	F/L
	Edi-0, Oy-0	1:18250605	A/C	SNP	Missense variant	Tolerated	E/A

## Data Availability

This published article and its Appendix A include all data generated or analyzed during this study.

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
