# Peer review of "Mapping Quantitative Trait Loci in Arabidopsis MAGIC Lines Uncovers Hormone-Responsive Genes Controlling Adventitious Root Development"

_plants, 2025, doi:10.3390/plants14111574_

Round 1
Reviewer 1 Report
Comments and Suggestions for Authors
The manuscript by B.A. López-Ruiz et al (Plants-3618752: “QTL Mapping Using Arabidopsis thaliana MAGIC Lines Identifies Candidate Genes Controlling Adventitious Root Development”) reports on the utilization of a QTL approach relying on a population of MAGIC Recombinant Inbred lines (RILs) to characterize the genetic architecture controlling natural variation in root system architecture. The authors characterized primary root length, lateral root number and length, and adventitious root and length, for 139 MAGIC lines. They then used QTL mapping to identify a region of chromosome 1 spanning 316 genes as significantly associated with the number of adventitious roots in this population. They analyzed this region, identifying a group of candidate genes and defining the genetic variation (SNPs) associated with the coding region of these candidates. Finally, they discussed the possible contribution of these candidates and their variants to the process of adventitious root formation.
This manuscript is well written and provides interesting and novel information on the genetic variation associated with root system architecture in this population of Arabidopsis MAGIC lines. However, this information remains of limited value without functional confirmation of contribution of these candidates, or at least one of them, to trait variation.
- The focus on SNPs disrupting the coding potential of these genes overlooks the fact that a substantial fraction of the phenotypic variation affecting quantitative traits in Arabidopsis (and plants in general) is associated with expression changes rather than alterations in the coding potential of the genes. Some attention should be given to expression differences between high- and low-adventitious rooting lines for genes within the mapped interval. Such an analysis could take advantage of some of the recent (sc)RNAseq analyses of adventitious root development to determine if identified DEGs in those studies map within this region of chromosome 1, as a first step to identify candidates (for instance);
- The mapping of SNPs within the coding region of candidate genes, along with their assignment to specific founder lines for the MAGIC population, is interesting, but incomplete. It seems that associating each discussed SNP with RILs of high- vs low-ARN values and also providing information on the ARN value of identified founding accessions, would be needed to better evaluate the significance of the association. Indeed, the data shown in figure S1 indicate higher complexity than what one would expect for a simple haplotype shared by all high- vs low-ARN accessions;
- More specific comments follow:
- Figure 1. Do the dots distributed above each box correspond to outliers? If they do, how were they identified (as the whiskers are indicated as extending to the maximum and minimum values)? Usually, if a graph shows outliers, the whiskers will represent a range of values defined by 1.5 times the interquartile range. Please specify what you represented in the legend;
- Also, what is the color code associated with the boxes in Figure 1 (which seems to follow a gradient from mainly green at the left to mainly red at the right)? Please provide an explanation for it in the legend (or as a color-scale bar next to the graphs);
- Figure 2. Please define the test that was used to determine the significance of correlation values between parameters (either in the legend, or in Materials and Methods);
- Table 1. Please define what you mean by “tolerated” or “deleterious”. Which parameters were used to define that effect on the product? This should be carefully explained within the table legend or in Materials and Methods;
- Figure 4. Please indicate the location of conserved domains I, II, III and IV in IAA/18;
- A supplemental Table listing the SNPs present along the significant segment of chromosome 1 (along with positioning of the genes along the segment), and defining the SNP alleles for all RILs, themselves sorted based on their ARN values, would be useful as it would provide an unambiguous/unbiased database summarizing all data, allowing a more complete interpretation of the results.
Author Response
REVIEWER 1
The manuscript by B.A. López-Ruiz et al (Plants-3618752: “QTL Mapping Using Arabidopsis thaliana MAGIC Lines Identifies Candidate Genes Controlling Adventitious Root Development”) reports on the utilization of a QTL approach relying on a population of MAGIC Recombinant Inbred lines (RILs) to characterize the genetic architecture controlling natural variation in root system architecture. The authors characterized primary root length, lateral root number and length, and adventitious root and length, for 139 MAGIC lines. They then used QTL mapping to identify a region of chromosome 1 spanning 316 genes as significantly associated with the number of adventitious roots in this population. They analyzed this region, identifying a group of candidate genes and defining the genetic variation (SNPs) associated with the coding region of these candidates. Finally, they discussed the possible contribution of these candidates and their variants to the process of adventitious root formation.
This manuscript is well written and provides interesting and novel information on the genetic variation associated with root system architecture in this population of Arabidopsis MAGIC lines. However, this information remains of limited value without functional confirmation of contribution of these candidates, or at least one of them, to trait variation.
Response. Thank you for your positive feedback on our work. We understand that functional validation is the best way to ensure the involvement of candidate genes in the target traits. This can be achieved by developing introgression lines that carry the introgressed QTL region on a consistent background or through targeted mutagenesis. However, due to time constraints, we decided to focus only on identifying candidate genes and, in the future, validate some of them in a more refined manner.
The focus on SNPs disrupting the coding potential of these genes overlooks the fact that a substantial fraction of the phenotypic variation affecting quantitative traits in Arabidopsis (and plants in general) is associated with expression changes rather than alterations in the coding potential of the genes. Some attention should be given to expression differences between high- and low-adventitious rooting lines for genes within the mapped interval. Such an analysis could take advantage of some of the recent (sc)RNAseq analyses of adventitious root development to determine if identified DEGs in those studies map within this region of chromosome 1, as a first step to identify candidates (for instance).
Response. We appreciate the reviewer’s insightful comment about the potential role of regulatory variation in explaining the observed phenotypic differences. We agree that differences in gene expression are a crucial source of phenotypic variation in Arabidopsis. However, our initial focus was on coding SNPs because they have clear interpretability in the MAGIC Lines. To address the reviewer comment, we have now examined published RNA-seq datasets related to adventitious root development (Liu et al., 2022). In the supplementary material, we have included a heatmap that shows the common genes found in the QTL interval of our study and those identified in a time-lapse analysis of adventitious root formation. Notably, we found that 177 out of 316 genes that we detected in the QTL peak were differentially expressed in these transcriptomes. Some of these genes overlap with the candidate genes we proposed (TOR, IAA18, Hydroxyproline-rich glycoprotein family protein, and GA2ox7). We have incorporated this information into the main text in the methods and results sections. While a comprehensive transcriptomic analysis of our specific lines is beyond the scope of this study, this integrative approach serves as a preliminary step to prioritize regulatory candidates for future investigation.
The mapping of SNPs within the coding region of candidate genes, along with their assignment to specific founder lines for the MAGIC population, is interesting, but incomplete. It seems that associating each discussed SNP with RILs of high- vs low-ARN values and also providing information on the ARN value of identified founding accessions, would be needed to better evaluate the significance of the association. Indeed, the data shown in figure S1 indicate higher complexity than what one would expect for a simple haplotype shared by all high- vs low-ARN accessions;
Response. We recognize that genotyping the MAGIC lines and associating the SNPs with the phenotype is the most effective method for establishing these associations. However, the available data from Kover et al. (2009) included only 1,536 SNPs across all the founders and MAGIC lines; of these, only 275 are in chromosome 1. As a result, we decided to utilize the database of founder lines, which is part of the 1001 Genomes Project and contains over 250,000 SNPs.
To ensure that we incorporate information about the polymorphisms of the MAGIC lines, we analyzed the SNPs provided by Kover et al. and searched for common SNPs among those identified in the 316 genes analyzed in our QTL study. We mapped the polymorphisms and discovered that only three SNPs are associated with the following genes: AT1G50300, AT1G51140, and AT1G52120. These genes encode a TBP-associated factor 15, a basic helix-loop-helix-type transcription factor involved in photoperiodic flowering, and a Mannose-binding lectin superfamily protein. None of these genes has been functionally validated for their role in adventitious root development.
To illustrate the differences in polymorphisms between the founder lines and the MAGIC lines we employed, we have added a new figure (Figure 4).
More specific comments follow:
Figure 1. Do the dots distributed above each box correspond to outliers? If they do, how were they identified (as the whiskers are indicated as extending to the maximum and minimum values)? Usually, if a graph shows outliers, the whiskers will represent a range of values defined by 1.5 times the interquartile range. Please specify what you represented in the legend;
Response. Thank you for your suggestions. Yes, the dots above each box represent outliers. These outliers were identified as data points that fall outside the range defined by the whiskers, using the default settings of the ggplot2 function in R. This information is included in the caption for Figure 1.
Also, what is the color code associated with the boxes in Figure 1 (which seems to follow a gradient from mainly green at the left to mainly red at the right)? Please provide an explanation for it in the legend (or as a color-scale bar next to the graphs).
Response. The color associated with each box in Figure 1 corresponds to individual MAGIC lines and was assigned based on their increasing order (left to right) using a manually defined color palette. Although the colors appear to follow a gradient, this is a visual effect of using a continuous color palette to distinguish categories and help visually differentiate the MAGIC lines. We have updated the text to clarify that the colors of the boxes are specific to each MAGIC line.
Figure 2. Please define the test that was used to determine the significance of correlation values between parameters (either in the legend, or in Materials and Methods);
Response. The significance of the correlation shown in the correlogram we created using the ggpairs() plot from the GGally package, as indicated by the Spearman correlation coefficient. This choice is appropriate because our variables are not normally distributed and contain outliers. We have included this information in both the Materials and Methods section and in the figure caption.
Table 1. Please define what you mean by “tolerated” or “deleterious”. Which parameters were used to define that effect on the product? This should be carefully explained within the table legend or in Materials and Methods;
Response. Thank you for your recommendation. According to the Variant Effect Predictor from Ensembl, the prediction of the effect of an amino acid substitution is likely to influence protein function based on sequence homology and the physicochemical similarity between the alternate amino acids. Each amino acid substitution is assigned a score and a qualitative prediction: either 'tolerated' or 'deleterious.' The qualitative prediction is based on this score, where substitutions with a score less than 0.05 are categorized as "deleterious," while all other substitutions are labeled as "tolerated." We have added this information to the materials and methods section and to the Table title.
Figure 4. Please indicate the location of conserved domains I, II, III and IV in IAA/18;
Response. Thank you for your suggestion. We have added domains I-IV to Figure 4. According to UniProt, IAA18 has an EAR domain, which consists of domain I (amino acids 42-46) and domain II, characterized by the sequence GWPPV. The domain III is KVDLSAHNSYEQLSFTVDK (164-182), and the domain IV has the following sequence: QRDFPSSIEDEKPITGLLDGNGEYTLTYEDNEGDKMLVGDVPWQMFVSSVKRLRVI.
A supplemental Table listing the SNPs present along the significant segment of chromosome 1 (along with positioning of the genes along the segment), and defining the SNP alleles for all RILs, themselves sorted based on their ARN values, would be useful as it would provide an unambiguous/unbiased database summarizing all data, allowing a more complete interpretation of the results.
Response. Thank you for your recommendation. We have added supplementary files containing the SNPs for the candidate genes, with each gene having approximately 150 to 400 SNPs (Table S5-S9). Additionally, we have included another table detailing the polymorphisms of the MAGIC lines and the associated genes that were mapped. Among these, only three SNPs are common between the 316 genes we have and the polymorphisms available for the RILS (Table S3). We have highlighted these SNPs in the table.
Reviewer 2 Report
Comments and Suggestions for Authors
I go through the manuscript plants-3618752 and find it very interesting however, it lacks scientific writeup and data representation. Here are some suggestions that may potentially improve the manuscript to reach the level of publication. Firstly, the current title is overly descriptive, and lacks focus on the key scientific conclusion. Forexample: Genomic Regions and Hormonal Pathways Underlying Adventitious Root Development in Arabidopsis thaliana MAGIC Populations. The abstract lack logical flow (e.g., methods/results/conclusions are muddled). Conclusions are vague and fail to highlight novel insights. The figures are low resolution (e.g., axis labels in Figure 2B are unreadable) etc.,. Inconsistent formatting (e.g., font sizes vary between panels). I suggest regenerate figures in vector formats (e.g., PDF/SVG) for scalability. Standardize font sizes (axes: 10–12 pt; labels: 14–16 pt). Add insets to magnify critical regions (e.g., QTL peaks overlapping candidate genes). In line 14 Arabidopsis thaliana is not italicized. Italicize the species name throughout the text (e.g., "Arabidopsis thaliana MAGIC lines…). Please make corrections to the keywords, which should be kept in ascending alphabetic order. I see many many grammer and typos which need tio be thouroughly corrected. These are just few examples: In Line 89: The MAGIC lines was grown…, Line 154: This results suggests…, Line 212: Further research is required to fully understand. These were just few examples. I suggest make corrections to the english grammer thouroughly and the revised version need to be approved from a native english speaker. In the whole manuscript please keep the all the genes names italic and the protein names should be kept straight. The materials and methods section are overly technical descriptions (e.g., "QTL mapping was performed using R/qtl2 with a 1.5 LOD threshold. Missing critical details (e.g., growth medium composition, imaging parameters). Plant Growth: Seeds were stratified for 48h at 4°C, then grown on ½ MS medium (pH 5.8) under 16h light/8h dark at 22°C. Imaging: Roots were imaged at 5 DAG using a Nikon SMZ25 stereomicroscope (10x magnification, 10 µm resolution). QTL Analysis: We used R/qtl2 (v1.6.4) with a 1.5 LOD threshold, permuting genotypes 10,000 times to establish significance. Provide a step-by-step workflow as a supplemental table for reproducibility. The conclusion section is overly broad statements (e.g., "This study advances our understanding of root biology"). Fails to link results to mechanistic hypotheses. I see many mistakes in the references section, such as typos mistakes in the titles, journal names and page number. etc. please double cross check the references section. The plagiarism is very high about 79% because th manuscript is uploaded in Research Square website. Please deal this issue.
Overall, I think the manuscript has potential to be published. However this manuscript needs thourough revision.
I see many many grammer and typos which need tio be thouroughly corrected. These are just few examples: In Line 89: The MAGIC lines was grown…, Line 154: This results suggests…, Line 212: Further research is required to fully understand. These were just few examples. I suggest make corrections to the english grammer thouroughly and the revised version need to be approved from a native english speaker.
Author Response
REVIEWER 2
I go through the manuscript plants-3618752 and find it very interesting however, it lacks scientific writeup and data representation. Here are some suggestions that may potentially improve the manuscript to reach the level of publication.
Firstly, the current title is overly descriptive, and lacks focus on the key scientific conclusion. For example: Genomic Regions and Hormonal Pathways Underlying Adventitious Root Development in Arabidopsis thaliana MAGIC Populations.
Response. Thank you for your suggestion. We have updated the title to include genes related to hormone response. Mapping Quantitative Trait Loci in Arabidopsis MAGIC Lines uncovers hormone-responsive genes controlling adventitious root development.
The abstract lack logical flow (e.g., methods/results/conclusions are muddled).
Response. We have revised the abstract's structure to include an introduction, methodology, results, and conclusions.
Conclusions are vague and fail to highlight novel insights.
Response. We have added the novel results from our study, including the possible roles of hormone-related genes identified in the QTL peak.
The figures are low resolution (e.g., axis labels in Figure 2B are unreadable) etc.,.Inconsistent formatting (e.g., font sizes vary between panels). I suggest regenerate figures in vector formats (e.g., PDF/SVG) for scalability. Standardize font sizes (axes: 10–12 pt; labels: 14–16 pt). Add insets to magnify critical regions (e.g., QTL peaks overlapping candidate genes).
Response. We have enhanced the image quality by increasing the DPI. There is no Figure 2 with panels; it is a single image.
In line 14 Arabidopsis thaliana is not italicized. Italicize the species name throughout the text (e.g., "Arabidopsis thaliana MAGIC lines…).
Response. Thank you for the suggestion. We have italicized all the species.
Please make corrections to the keywords, which should be kept in ascending alphabetic order.
Response. We have changes that include the keyword in alphabetical order.
I see many many grammer and typos which need tio be thouroughly corrected. These are just few examples: In Line 89: The MAGIC lines was grown…, Line 154: This results suggests…, Line 212: Further research is required to fully understand. These were just few examples. I suggest make corrections to the english grammer thouroughly and the revised version need to be approved from a native english speaker.
Response. Thank you for your comments. However, the sentences you referred to—Line 89: "The MAGIC lines was grown…" and Line 154: "This results suggests…"—do not appear in our manuscript. It seems there may be a misunderstanding regarding the text that you mentioned. Nevertheless, we have thoroughly reviewed the manuscript, and Prof. Joshua Banta, a coauthor who is a native English speaker, has approved the final version.
In the whole manuscript please keep the all the genes names italic and the protein names should be kept straight.
Response. We have reviewed the manuscript thoroughly. Genes are in uppercase and italics, while proteins are only in uppercase.
The materials and methods section are overly technical descriptions (e.g., "QTL mapping was performed using R/qtl2 with a 1.5 LOD threshold. Missing critical details (e.g., growth medium composition, imaging parameters). Plant Growth: Seeds were stratified for 48h at 4°C, then grown on ½ MS medium (pH 5.8) under 16h light/8h dark at 22°C. Imaging: Roots were imaged at 5 DAG using a Nikon SMZ25 stereomicroscope (10x magnification, 10 µm resolution).
Response. We have provided details about the culture medium and the image digitization process.
QTL Analysis: We used R/qtl2 (v1.6.4) with a 1.5 LOD threshold, permuting genotypes 10,000 times to establish significance. Provide a step-by-step workflow as a supplemental table for reproducibility.
Response. We have added the Supplemental Table S10 with a step-by-step workflow for QTL mapping and gene overlay in Arabidopsis MAGIC lines
The conclusion section is overly broad statements (e.g., "This study advances our understanding of root biology"). Fails to link results to mechanistic hypotheses.
Response. Thank you for your comment. We have improved the conclusion and added the possible roles of these hormone-related genes in adventitious root formation.
I see many mistakes in the references section, such as typos mistakes in the titles, journal names and page number. etc. please double cross check the references section.
Response. Thank you for your observation. We have utilized Zotero as our reference management software to manage our bibliographic data. We have configured Zotero to align the references with the journal's requirements.
The plagiarism is very high about 79% because th manuscript is uploaded in Research Square website. Please deal this issue.
Response. Thanks for the observation. From the moment we submitted our manuscript to Plants, we declared that it was previously published in the platform Research Square as a preprint. To the best of our knowledge, it is not possible to remove a preprint from Research Square once it has been published and assigned a DOI.
Overall, I think the manuscript has potential to be published. However this manuscript needs thourough revision.
Response. Thank you for the suggestions. We have improved our manuscript based on your comments and those of the other reviewer. We have also carefully re-read the text and made the necessary corrections.
Comments on the Quality of English Language
I see many many grammer and typos which need tio be thouroughly corrected. These are just few examples: In Line 89: The MAGIC lines was grown…, Line 154: This results suggests…, Line 212: Further research is required to fully understand. These were just few examples. I suggest make corrections to the english grammer thouroughly and the revised version need to be approved from a native english speaker.
Response. We have thoroughly reviewed the manuscript, and a native-speaking author, Prof. Joshua Banta from the University of Texas at Tyler, has revised the final version.
Round 2
Reviewer 1 Report
Comments and Suggestions for Authors
The authors addressed most of the points I raised on the previous draft, and the manuscript has been improved in the process. However, the incorporation of published expression in data analysis supporting the contribution of a suggested list of candidates (TOR, IAA18, Hydroxyproline-rich glycoprotein family protein, and GA2ox7) begs for expression analysis of these candidates between lines and correlation studies (as suggested in my previous review).
Author Response
Reviewer 1
The authors addressed most of the points I raised on the previous draft, and the manuscript has been improved in the process. However, the incorporation of published expression in data analysis supporting the contribution of a suggested list of candidates (TOR, IAA18, Hydroxyproline-rich glycoprotein family protein, and GA2ox7) begs for expression analysis of these candidates between lines and correlation studies (as suggested in my previous review).
Response
Thank you for acknowledging our efforts to improve our article. We totally agree that we are only providing candidate genes and establishing associations with the existing literature and transcriptome studies, which we believe supports their candidacy. We agree with you that gene expression studies would be ideal to further support our candidate genes, and yet gene expression is not the only one study still to be done. Actually besides gene expression studies it would be ideal to do allele swaps, mutant analyses, graftings, etc. None of these evidences on their own will be the definitive proof of the candidate genes, so we consider that these approaches are the matter of follow-up research endeavors to be done either by our research group or the community. We apologize for not pursuing your request, acknowledging that it would add one more element but would be far from being totally conclusive for our current manuscript. The editorial times of Plants are tight, and this follow-up research will take months, or even years, to be accomplished. We hope to have your understanding that our findings need to be communicated so more research groups can get on board on the validation of candidate genes for this essential trait of adventitious root development.
Reviewer 2 Report
Comments and Suggestions for Authors
The authors satisfactory replied to the comments and made significant corrections to the manuscript. I accept and support the manuscript for publication.
Author Response
Reviewer 2
The authors satisfactory replied to the comments and made significant corrections to the manuscript. I accept and support the manuscript for publication.
Response. Thank you for your comments and suggestions that help us improve our article and support its publication.